# Surface Bioactivation of Polyether Ether Ketone (PEEK) by Sulfuric Acid and Piranha Solution: Influence of the Modification Route in Capacity for Inducing Cell Growth

**DOI:** 10.3390/biom11091260

**Published:** 2021-08-24

**Authors:** Flavia Suzany Ferreira dos Santos, Mariana Vieira, Henrique Nunes da Silva, Helena Tomás, Marcus Vinícius Lia Fook

**Affiliations:** 1Departament of Materials Engineering, Federal University of Campina Grande, Campina Grande 58429-900, PB, Brazil; flaviasuzanyfs@gmail.com (F.S.F.d.S.); henrique.nunes.silva.eng@gmail.com (H.N.d.S.); 2CQM—Centro de Química da Madeira, Universidade da Madeira, Campus da Penteada, 9020-105 Funchal, Portugal; mariana.vieira@staff.uma.pt (M.V.); lenat@staff.uma.pt (H.T.)

**Keywords:** piranha reagent, functionalization, sulfuric acid, PEEK, surface chemistry

## Abstract

The aim of this study was to promote bioactivity of the PEEK surface using sulfuric acid and piranha solution. PEEK was functionalized by a sulfuric acid treatment for 90 s and by piranha solution for 60 and 90 s. Chemical modification of the PEEK surface was evaluated by infrared spectroscopy, contact angle analysis, cytotoxicity, cell adhesion and proliferation. The spectroscopy characteristic band associated with sulfonation was observed in all treated samples. PEEK with piranha solution 60 s showed an increase in the intensity of the bands, which was even more significant for the longer treatment (90 s). The introduction of the sulfonic acid functional group reduced the contact angle. In cytotoxicity assays, for all treatments, the number of viable cells was higher when compared to those of untreated PEEK. PEEK treated with sulfuric acid and piranha solution for 60 s were the treatments that showed the highest percentage of cell viability with no statistically significant differences between them. The modified surfaces had a greater capacity for inducing cell growth, indicative of effective cell adhesion and proliferation. The proposed chemical modifications are promising for the functionalization of PEEK-based implants, as they were effective in promoting bioactivation of the PEEK surface and in stimulating cell growth and proliferation.

## 1. Introduction

The polymer polyether ether ketone (PEEK) has stood out as a promising material for orthopedic implants mainly due to its characteristics of biocompatibility, and mechanical and chemical resistance [1]. Moreover, PEEK has mechanical properties such as elastic modulus and tensile strength closer to those of human bones when compared to metal implants [2]. Thus, the PEEK polymer is known as an alternative biomaterial to implantable metal materials. However, it is biologically inert and therefore it is not suitable for applications requiring interactions between implant and bone tissues. These interactions are part of a process called osseointegration which is essential for bone regeneration. Without this condition implants may become loose or migrate, causing pain, deformity, or deficiency in the patient [3,4,5].

Therefore, there were various mechanisms proposed and developed in order to improve PEEK bioactivity and consequently, a better tissue-implant interface, using physical or chemical modifications [6,7,8,9,10,11].

The commonly used physical and chemical treatments of the PEEK surface are, respectively, plasma modifications and accelerated neutral atom beam; and sulfonation treatment and wet chemical modification [4]. Chemical modifications alter the chemical structure of the material surface to create a specific environment that provides a favorable cellular response [12]. Sulfonation has become a promising alternative in chemical surface modification of PEEK. It is quite effective to introduce the functionality of sulfonic acid into the PEEK chain making the polymer hydrophilic and increasing its bioactivity [13]. Concentrated sulfuric acid (98%) is the most commonly selected sulfonating agent [14,15].

Different approaches regarding the chemical modification of PEEK with sulfuric acid have been reported. Almasi, et al. [16] evaluated different immersion times of PEEK in sulfuric acid and observed that the SO_3_H group was introduced into the polymeric chain. Wang, et al. [17] evaluated the hydrophilicity and morphology of PEEK under different sulfonation conditions and observed an increase of the hydrophilic character of the modified surface compared to the unmodified one that can effectively improve the hydrophilicity of the surface of PEEK and introduce bioactive nano-topography for bone implant applications. Montero, et al. [18] evaluated the antimicrobial behavior of SPEEK (sulfonated PEEK) membranes for *Streptococcus mutans* and *Enterococcus faecalis* colonies and reported that the sulfonation process affected the growth of SPEEK biofilm, revealing a potential modification of PEEK that could improve its antimicrobial activity. Zhao, et al. [19] observed that the surface modification sulfonation technique improved cell adhesion and proliferation in PEEK, besides inducing apatite formation on the modified surface. Another viable alternative is thermal sulfonation. This may have a direct influence on the increase in degree of sulfonation and the ion exchange capacity, in addition to influencing other parameters such as water absorption and permeability due to presence of high content of sulfonic acid groups [20].

Another chemical modification technique that has been studied is the use of a solution of sulfuric acid with hydrogen peroxide (H_2_O_2_), called “piranha solution”. Using this strong corrosive and oxidizing agent, the reported effects are increased PEEK hydrophilicity and surface energy due to the increased amount of functional groups available by the synergistic action of their constituents [11,21,22,23,24,25,26]. Comparative studies of pre-treatments of dental PEEK resins with sulfuric acid and piranha solution have been reported [23,24]. However, to the best of our knowledge, only adherence and/or mechanical evaluations were performed in these. No approaches have been reported comparing the influence of process parameters (concentration and time) of these treatments on PEEK properties.

Thus, the objective of this study was to achieve bioactivity on the PEEK surface with different methods, using only sulfuric acid and piranha solution in different reaction times, and to evaluate the influence of the treatment on the physicochemical and biological properties of the modified PEEK surface.

## 2. Materials and Methods

### 2.1. Materials

Medical-grade PEEK (Victrex^®^ TecaPEEK Classix White) materials used in this study were provided by Ensinger (São Leopoldo, RS, Brazil). Disk samples with dimensions of ɸ20 × 8 mm were prepared for surface characterization, immersion tests and in vitro studies. Aluminum carbide sandpaper (Al4C3) (600#, 1200# and 2000#) was provided by Deerfos^®^ (Maringá, Brazil). Sulfuric acid 98% was obtained from Dinâmica^®^ (Sao Paulo, Brazil) and hydrogen peroxide 35% from Neon^®^ (Sao Paulo, Brazil).

### 2.2. Preparation of Functional Surface PEEK

The surface of the PEEK samples was prepared with Al_4_C_3_ sandpaper (600#, 1200# and 2000#) coupled to a metallographic polishing machine PLO2E Teclago^®^ (São Paulo, Brazil). After sanding, the samples were washed in an ultrasonic bath (Ultracleaner 1440 plus Unique, São Paulo, Brazil) with distilled water at 25 ± 1 °C for 10 min. Subsequently, the PEEK samples were oven-dried at 60 ± 0.5 °C for 20 min.

To achieve chemical modification of the surface, PEEK samples were submerged in sulfuric acid (98%) for 90 s and piranha solution (2:1 of H_2_SO_4_ (98%) for H_2_O_2_ (35%)) for 60 and 90 s. The respective codenames for each PEEK sample are described in Table 1. All treatments were performed at 25 ± 1 °C.

Afterwards, treated samples were washed repeatedly with distilled water at room temperature (25 ± 1 °C) for 10 min, and then washed in distilled water at 100 °C with pH verification, until a complete extraction of the residual sulfuric acid was achieved [18,19]. Subsequently, specimens were oven dried at 60 °C for 20 min.

### 2.3. Surface Characterizations

#### 2.3.1. Fourier Transform Infrared Spectroscopy (FTIR)

FTIR (Perkin Elmer Spectrophotometer Spectrum 400 (Waltham, MA, USA) in the absorbance mode was used to confirm the presence of sulfonated groups after the treatments, by analyzing the polymer functional groups of the treated PEEK samples and the untreated control samples. A scanning range in the average infrared region of 550–4000 cm^−1^ was performed at 25 ± 1 °C.

#### 2.3.2. Contact Angle Analysis

Contact angle measurements were carried out in a drop shape analyzer (Model DSA100S, KRÜSS^®^, Hamburg, Germany) to evaluate the surface hydrophilicity of the samples. At room temperature (22 ± 1 °C), a deionized water droplet was dropped onto each sample surface and pictures were taken by a camera after stabilization. Five parallel tests were carried out on the different areas of each sample. Based on the ASTM standard (D7334) [27], contact angle was measured within a period of less than 30 s after the drop.

#### 2.3.3. Cytotoxicity Assay

Cytotoxicity assays were performed using extracts from the in vitro degradation of the chemically modified PEEK samples. To obtain the extracts, samples were sterilized in an autoclave for 30 min, and after that, immersed in Dulbecco’s Modified Eagle’s Medium (DMEM) and left inside an incubator at 37 °C. The extracts containing the degradation products from the samples were then collected after 7 days. For the cytotoxicity studies, three different dilutions of each extract (1:10, 1:2 and 1:1) were prepared in DMEM, corresponding to three different extract concentrations (10%, 50% and 100%).

Human fibroblast cells BJh-TERT (ATCC^®^CRL-2522) were selected for the cytotoxicity study based on the ISO 10993-5:2009 standard (ISO, 2009). Cytotoxicity was evaluated by determining the cell viability using the resazurin reduction method. Cell viability was determined in relation to a positive control (100%), which consisted of BJh-TERT cells grown in DMEM medium with no added extract. Cells were seeded onto 96-well plates with a cell density of 10,000 cells/well, followed by incubation for 24 h at 37 °C/5% CO_2_. Extracts with all different concentrations for each sample were added to the wells, in triplicates, and further incubated for 24 h at 37 °C/5% CO_2_. Every two days, the preparation (DMEM + extract) was changed, until a final period of seven days for evaluation. After this period, the old solution was changed into a new one, containing 10% resazurin, followed by incubation for 3 h at 37 °C/5% CO_2_. Then, 100 μL of the medium was used to measure the fluorescence of the resorufin in a Victor^3^ 1420 PerkinElmer microplate reader (λ_exc_ = 530 nm; λ_em_ = 590 nm).

#### 2.3.4. Cell Adhesion and Proliferation

Samples were sterilized in an autoclave for 30 min. To look at cell adhesion and proliferation, a direct contact method between material and cells was used. Human fibroblasts cells BJh-TERT (ATCC^®^CRL-2522) were selected for the assay, which was done thrice. Cells were seeded on each sample, as triplicates, in 12-well plates at a density of 20.000 cells/well in DMEM (with 10% FBS + 1% antibiotic/antimycotic) and cultured for 24 h, 3 and 7 days at 37 °C/5% CO_2_. Afterwards, samples with cells were washed twice with PBS and fixed with 3.7% formaldehyde. Nuclei were stained with DAPI (Sigma Aldrich, St. Louis, MO, USA), the cytoskeleton protein F-actin was stained with Phalloidin (Sigma Aldrich) and cells were examined by fluorescence microscopy (in Nikon Eclipse TE 2000-E equipment).

The evaluation of the specimens surface morphology after cell adhesion was also done by scanning electron microscopy (SEM), using a Hitachi model TM-1000 (Chiyoda, Tokyo, Japan) electron microscope with a maximum magnification of 10,000×, depth of focus of 1 mm, resolution of 30 nm, 15 KV, low vacuum and varied pressure (1 to 270 Pa), with a metallic coating. For the application of this technique, magnifications of 1500× and 3000× were used.

#### 2.3.5. Statistical Analysis

One-way ANOVA analysis was performed using Minitab version 19.1 software, and significant differences between conditions were determined by Tukey’s test with a statistical significance level of 95%.

## 3. Results and Discussion

### 3.1. Fourier Transform Infrared Spectroscopy (FTIR)

Figure 1a shows the FTIR spectra of PEEK, SPEEK, PS-PEEK-60 and PS-PEEK-90 samples. The graph region from 1800 to 550 cm^−1^ is enlarged to highlight the differences between PEEK and the respective treatments (Figure 1b).

The spectrum of the untreated PEEK sample showed all characteristic bands [28,29]: band at 1221 cm^−1^, referring to the asymmetric stretching of the C-O aromatic ether structure [30]; bands at 1650, 1490, 926, 1157, 1185 cm^−1^ corresponding to the diphenylketone linkage [1]; bands at 1306 cm^−1^ and 1278 cm^−1^ associated with the C=O group of the ketone linkage and the resonance of the di-phenyl ether group, respectively; and the C=O stretch corresponding to the benzophenone units that are located in the bands at 1594 and 1648 cm^−1^ [19].

The characteristic band associated to sulfonation was observed in all treated PEEK samples, this being highlighted by the presence of sulfonic acid groups at 3440 and 1050 cm^−1^ [18]. The wide band at 3440 cm^−1^ is attributed to the OH- vibration of the hydroxyl group of the sulfonic acid functional group (SO_3_H) (occurring exclusively in the oxy-1,4-henyleneoxy rings) [RS(=O)2-OH] [7]. The vibration at 1050 cm^−1^ corresponds to the symmetrical stretching of the S=O band [31,32].

In the spectra of PS-PEEK-60 an increase in the intensity of the band at 3440 cm^−1^ when compared to the spectrum of the SPEEK sample, was observed. This effect indicates the oxidation of PEEK which occurs through the oxygen released during the reaction of the hydrogen peroxide with sulfuric acid that reacts directly with the aromatic ring of the benzene group [13,33]. This leads to an increase in the surface polarity and aromatic ring opening, resulting in a greater quantity of functional groups available to interact with the physiological environment when implanted [11,23]. The disappearance of the vibration bands of the CH=CH bond at 744 cm^−1^ and 865 cm^−1^, as well as the appearance of bands at 1445 cm^−1^, corresponding to the angular deformation of -(CH_2_) out of the plan, were also observed [13]. These results also suggest the opening of the aromatic ring of PEEK [11,23].

The PS-PEEK-90 spectrum showed even more intense band variations compared to the other samples. In addition to the disappearance of the 865 cm^−1^ and 744 cm^−1^ vibration bands, already observed in the PS-PEEK-60 spectrum, it was also noticeable the reduction of the band intensity at 965 cm^−1^ of the CH=CH. The band at 673 cm^−1^, which is related to the angular deformation of the aromatic ring, also disappeared reinforcing the possibility of the breaking of the aromatic ring due to the increase in reaction time.

During the oxidation reaction of the piranha solution, various oxidants (O, H_3_O, HO, HSO_4_ and H_2_SO_5_) are formed transiently in a very short time resulting in H_2_SO_5_ (persulfuric acid or Caro’s acid) and H_2_O in steady-state with very low pH value. Caro´s acid is unstable and decomposes to form hydroxyl radicals that are responsible for its oxidizing power [34,35]. Thus, these radicals may be responsible for the severe chemical modification on the PEEK surface treated with piranha solution, a factor observed mainly in the 3440 cm^−1^ band corresponding to the -OH bond.

### 3.2. Contact Angle Analysis

Figure 2 shows the contact angle values of PEEK with and without all treatments. Contact angle is an important characteristic of biomaterials because it is related to interactions between the functional groups present on the surface of the material with the tissue adjacent to the implant.

Untreated PEEK samples presented a water contact angle of 78.6 ± 3.1°, similar to the one reported in the literature [18]. SPEEK samples presented a contact angle of 67.2 ± 3.1°, which is into the expected range for samples treated with sulfuric acid [7,16,17]. Treatments with piranha solution resulted in greater hydrophilicity compared to PEEK and SPEEK samples. In the PS-PEEK-60 sample there was a reduction of the angle of 19% when compared to PEEK, from 78.6 ± 3.1° to 63.5 ± 2.0°. In the PS-PEEK-90 treatment, the reduction was of 44% when compared to PEEK, from 78.6 ± 3.1° to 43.8 ± 2.8°. This was the treatment which made the surface more hydrophilic. According to Tukey’s test, only SPEEK and PS-PEEK-60 samples do not differ statistically in the contact angle values (*p*-value = 0.068), similarly to what was observed in the FTIR spectra of these samples.

Introduction of a hydrophilic HSO_3_ functional group on SPEEK and PS-PEEK surfaces, confirmed by FTIR analysis, has noticeably reduced the contact angle when compared to that of PEEK. Furthermore, an increase in immersion time in piranha solution caused an increase in the quantity of this functional group, implying a larger decrease of PS-PEEK-90 contact angle when compared to other treatments.

These observed effects of the chemical modifications are promising for decreasing the bioinert character of the PEEK surface. Indeed, a surface with greater hydrophilicity has been shown to be favorable for increased cell interaction, proliferation and adhesion, which could accelerate bone healing, osteogenic capacity and implant osteointegration [19,36,37,38,39].

### 3.3. Cytotoxicity Assay

Figure 3 presents the cytotoxicity results. It was observed, based on the Tukey’s test, that among the tested extract concentrations (10%, 50%, 100%) there was no significant difference (*p*-value > 0.05) in the cell viability percentage. In all cases, it was observed that treatments did not affect cell viability when compared to the PEEK untreated sample. All of them presented BJh-TERT fibroblasts viability above the established value for cytotoxicity by the ISO 10993-5 2009 standard (70%) (represented as a red line in Figure 3). Tukey’s test showed no statistical difference in cell viability results between the following treatments pairs: PEEK and PS-PEEK-90 (*p*-value = 0.655), and SPEEK and PS-PEEK-60 (*p*-value = 0.940).

For all treatments, the number of viable cells was higher when compared to untreated PEEK. SPEEK and PS-PEEK-60 samples showed the highest percentage of cell viability with no statistically significant difference between the extract concentrations tested (*p*-value = 0.940). It seems that the simultaneous use of hydrogen peroxide and sulfuric acid leads to a synergistic reaction favorable to cell’s development and viability. This is an important observation regarding the potential of this modified polymer for biomedical applications.

In PS-PEEK-90, a slight decrease in cell viability was observed when compared to SPEEK and PS-PEEK-60 treatments. The slight decrease in cell viability caused by this treatment may be related the greater number of chemical changes on the surface, as mentioned before when analyzing FTIR results, suggesting a greater number of OH radicals were responsible for the oxidizing power on the surface, however, it still showed values within the norm.

### 3.4. Cell Adhesion and Proliferation

After staining the nuclei of viable cells with DAPI, cell adhesion (after 24 h) and proliferation (after 3 and 7 days) over the surfaces of treated and untreated PEEK samples were observed by fluorescence microscopy (Figure 4).

After 24 h of direct contact between cells and the surface of the samples, it was observed that in all cases there was adhesion since cells remained intact, healthy and initially adhered well to the surfaces. Qualitatively, the SPEEK surface showed a greater number of adherent cells during the first 24 h (Figure 4b).

After 3 days (Figure 4e–h), it was possible to identify cell proliferation in all treated surfaces when compared to untreated PEEK. The largest areas covered by cells were observed on the surfaces with the SPEEK and PS-PEEK-60 treatments. This indicated that the chemical modifications of the surface promoted by these treatments provided functional groups that allowed a better cell attachment and stimulated cell proliferation. Interestingly, the biological assays and the physical-chemical tests previously described (FTIR and contact angle) show that the treatment with piranha solution for 60 s (PS-PEEK-60) achieves similar results to the treatment with only sulfuric acid for 90 s (SPEEK).

At the end of 7 days, cell proliferation was also observed after DAPI and phalloidin staining by fluorescence microscopy (Figure 5) and by SEM (Figure 6). The PEEK surface (Figure 4i and Figure 5a) did not show any major changes in the amount of adherent cells when compared to 24 h and 3 days of growth (Figure 4a,e). Although cells appear to remain in good shape, PEEK’s untreated surface did not seem to stimulate their growth.

Regarding the proposed treatments, it was observed that the modified surfaces had a greater capacity for inducing cell growth since there was a visible increase in the number of cells over time (Figure 4j–l). SPEEK (Figure 4j and Figure 5b) and PS-PEEK-60 (Figure 4k and Figure 5c) treatments showed better results when compared to the PS-PEEK-90 treatment (Figure 4l and Figure 5d).

In the case of PS-PEEK-90, the lower number of cells adherent to the modified surface can be justified by the greater functionalization of the surface due to increased reaction time in piranha solution which affected cell proliferation.

In Figure 6, SEM micrographs before (Figure 6a,d,g,j) and after (Figure 6b,e,h,k) cell adhesion and proliferation are shown, in untreated and treated PEEK surfaces. It is possible to observe that fibroblast cells showed a healthy fusiform aspect with various cell extensions (as indicated by arrows in Figure 6b,e,h,k). Similar results were found by [19,39]. When we enlarge SEM micrographs, we can see this format more clearly (Figure 6c,f,i,l).

Although cell morphology when grown on the treated surfaces is similar to that when grown on untreated PEEK, the number of cells adherent to SPEEK and PS-PEEK-60 was higher (Figure 6e,h).

The greatest surface covering of cells that was observed, originating from effective cell adhesion and proliferation, is an important factor when considering polymer applications in the biomedical area, especially in regions where greater interaction between implant and tissue is needed. Initial cell adhesion is generally responsible for subsequent cell proper functions and eventual tissue integration, and cell proliferation is correlated with new bone formation. Therefore, cell adhesion and proliferation are likely to favor the production of bone tissue around the implant and a stronger bone-implant bond is expected in vivo [19]. Past studies have shown that the surface modification of PEEK may have a significant influence on its topography and chemistry, enhancing its biological activity properties (Gan, et al. [40]. The present studies also show evidences of this behavior, revealing that an adequate acid treatment of the PEEK surface can make it more cytocompatible. However, further studies are needed to confirm the obtained results, namely performing in vitro tests with cells that better model the material-bone interaction (like osteoblastic cells) and, in addition, testing the materials in vivo using animal models.

## 4. Conclusions

Poly (ether-ether-ketone) (PEEK) surface was bioactivated using two different methods, with sulfuric acid and piranha solution over two different time periods. In all treatments, the functionalization of PEEK by sulfonation was confirmed by the addition of hydroxyl groups (-OH) from the sulfuric acid functional group, and the water contact angle was decreased. The functionalization was more severe for samples treated with piranha solution for 90 s, resulting in the smallest water contact angle. Cytotoxicity assays revealed that extracts from all treated PEEK samples resulted in cell viability greater than that of the extract from the untreated sample. Cell adhesion and proliferation were lower in the untreated sample and the one treated with piranha solution for 90 s, due to their bioinert surface and greater surface functionalization, respectively. Cells remained healthy and with typical morphology, indicating that the treatments were effective. The proposed chemical modifications were successful in promoting bioactivation of the PEEK surface, which makes them promising for functionalization of PEEK-based implants.

## Figures and Tables

**Figure 1 biomolecules-11-01260-f001:**
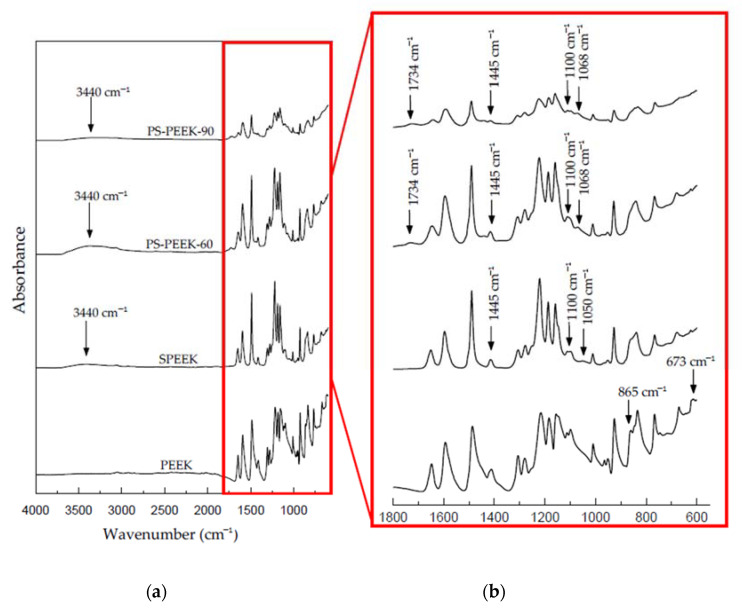
Comparative FTIR spectra of PEEK, SPEEK, PS-PEEK-60 and PS-PEEK-90 samples samples (**a**) and enlargement of the graph in the region from 1800 to 550 cm^−1^ to highlight the differences between PEEK and the respective treatments (**b**).

**Figure 2 biomolecules-11-01260-f002:**
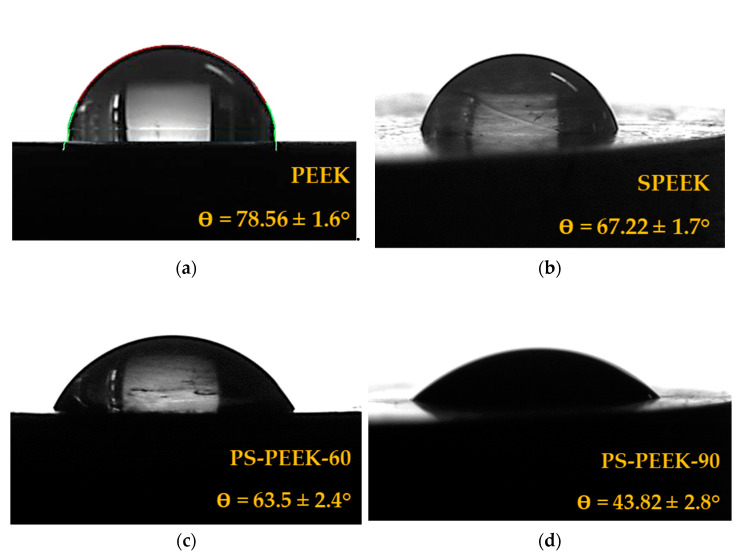
Contact angle of (**a**) PEEK, (**b**) SPEEK, (**c**) PS-PEEK-60 and (**d**) PS-PEEK-90 samples.

**Figure 3 biomolecules-11-01260-f003:**
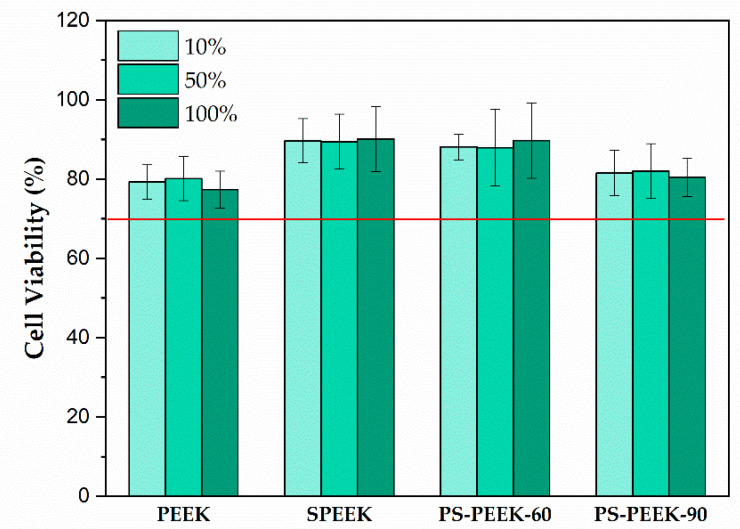
Evaluation of cell viability percentage of PEEK, SPEEK, PS-PEEK-60 and PS-PEEK-90 extracts in contact with cells and minimum percentage (red line) that corresponds to the value established by the ISO 10993-5 2009 standard.

**Figure 4 biomolecules-11-01260-f004:**
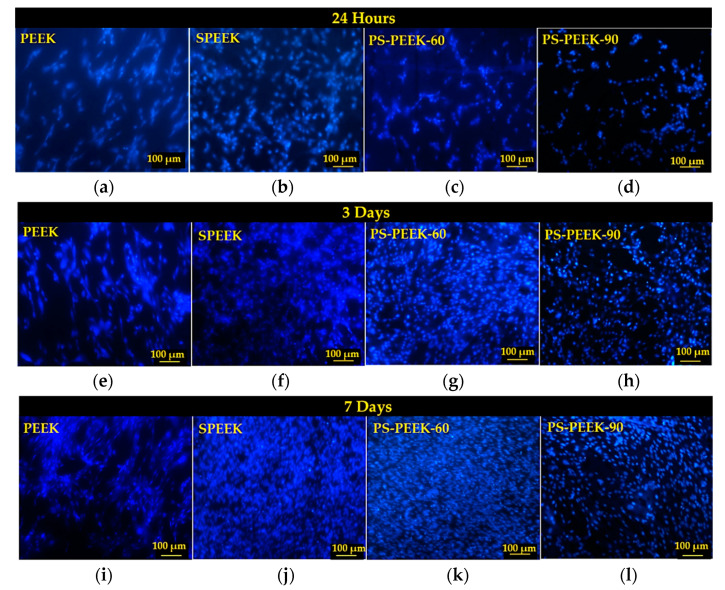
Cells stained with DAPI adhered to the surfaces of PEEK (**a**,**e**,**i**), SPEEK (**b**,**f**,**j**), PS-PEEK-60 (**c**,**g**,**k**) and PS-PEEK-90 (**d**,**h**,**l**) samples after 24 h, 3 and 7 days of growth.

**Figure 5 biomolecules-11-01260-f005:**
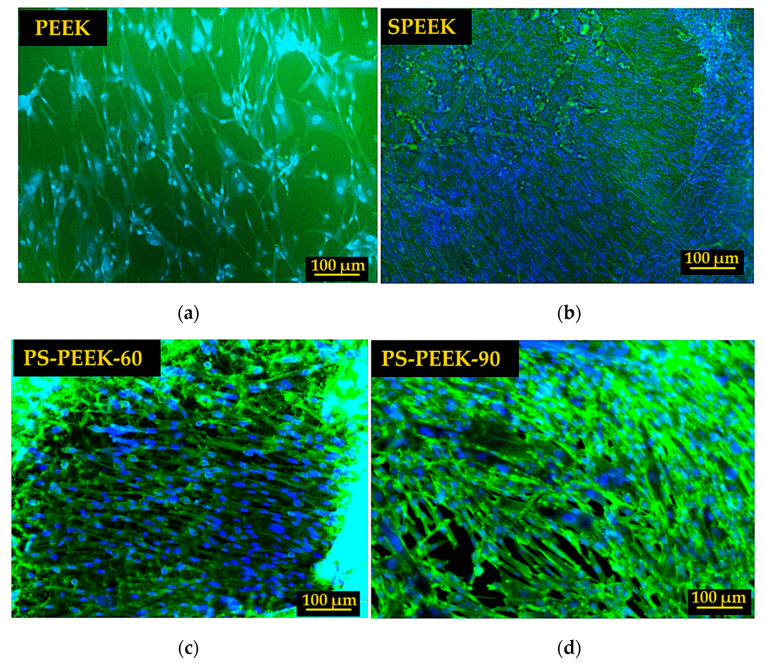
Cells stained with DAPI (blue) and phalloidin (green) adhered to the surfaces of PEEK (**a**), SPEEK (**b**), PS-PEEK-60 (**c**) and PS-PEEK-90 (**d**) samples after 7 days of growth.

**Figure 6 biomolecules-11-01260-f006:**
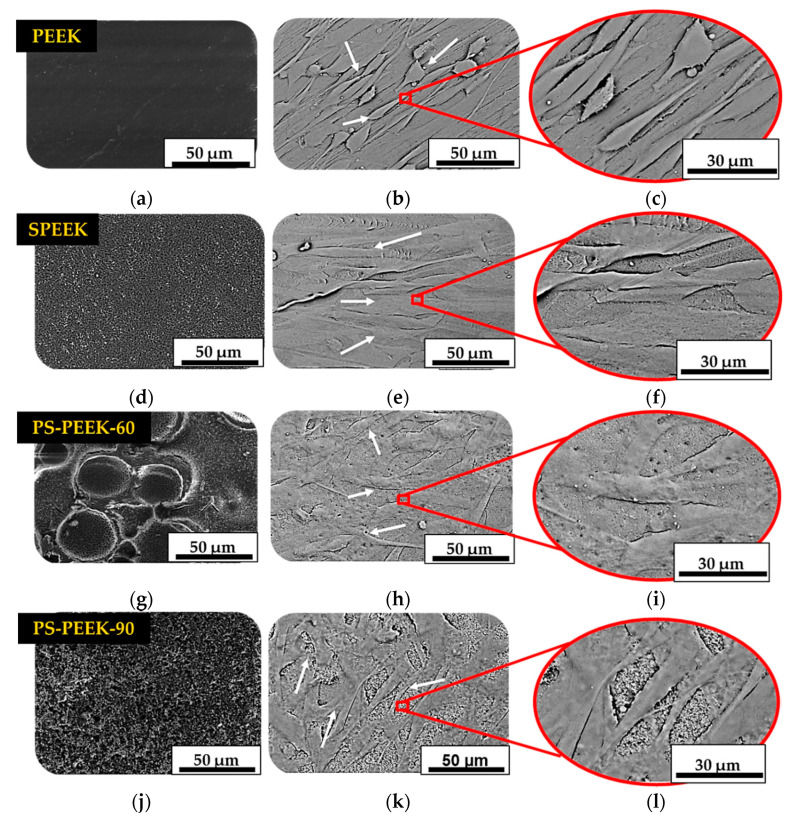
SEM micrographs of PEEK (**a**–**c**), SPEEK (**d**–**f**), PS-PEEK-60 (**g**–**i**) and PS-PEEK-90 (**j**–**l**) surfaces before and after cell adhesion and proliferation after 7 days.

**Table 1 biomolecules-11-01260-t001:** Test parameters and sample identification.

Treatment	Immersion Time(s)	Code Names
Sulfuric Acid H_2_SO_4_ (98%)	90	SPEEK
Piranha Solution 2:1 *w/v* H_2_SO_4_ (98%):H_2_O_2_ (35%)	60	PS-PEEK-60
Piranha Solution 2:1 *w/v* H_2_SO_4_ (98%):H_2_O_2_ (35%)	90	PS-PEEK-90

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
