# Peer review of "Surface Bioactivation of Polyether Ether Ketone (PEEK) by Sulfuric Acid and Piranha Solution: Influence of the Modification Route in Capacity for Inducing Cell Growth"

_biomolecules, 2021, doi:10.3390/biom11091260_

Round 1

Reviewer 1 Report

This is an interesting paper on the surface bioactivation of PEEK by sulfuric acid and piranha solution. The results are well explained and significant.

I had a minor concern regarding the novelty of the work, but in fact, the PEEK sulfonation with sulfuric acid and piranha solution has been studied to a very limited extent before (e.g. Stawarczyk, B.; Jordan, P.; Schmidlin, P. R.; Roos, M.; Eichberger, M.; Gernet, W.; Keul, C., The Journal of Prosthetic Dentistry 2014, 112, (5), 1278-1288 and Uhrenbacher, J.; Schmidlin, P. R.; Keul, C.; Eichberger, M.; Roos, M.; Gernet, W.; Stawarczyk, B., J Prosthet Dent 2014, 112, (6), 1489-97, both are cited in this manuscript). In addition to that, also biological properties including cell adhesion and proliferation as well as cytotoxicity assays are added in this study (which were not presented in previous studies). Hence, in my opinion, the manuscript deserves publication.

Specified comments:

  1. Abstract: Line 26-27 – typo error “inducing cell growth cells”,
  2. Introduction section lacks information about other possibilities of PEEK sulfonation, e.g thermal sulfonation in presence of sulphides [T. Moskalewicz, A. Kruk, M. Sitarz, A. Kopia, J. Electrochem. Soc. 166 (2019) D151-D161]. The authors discuss only chemical sulfonation with concentrated sulfuric acid. I think another alternative should also be mentioned. It would be valuable for the reader.
    There are also other studies on chemical sulfonation of PEEK, particularly the influence of the -SO3H group on its hydrophilicity: e.g. M.H.D. Othman, A. F. Ismail, A. Mustafa, Malaysian Polym. J. 2 (2007) 10 -28; W. Wang, C.J. Luo, J. Huang, M. Edirisinghe, J. R. Soc. Interface 16 (2019) 20180955, which could be also discussed for completeness. Please expand the introduction by discussing previous relevant papers.
  1. Results: What was the structure of the virgin PEEK, amorphous or semi-crystalline? How did the structure change after sulfonation? Please add a suitable discussion to the manuscript.

Author Response

Abstract: Line 26-27 – This correction was carried out.

Introduction: According to literature, PEEK is a polymer that has a high chemical resistance to solvents, except to concentrated sulfuric acid (97, 98%). This was selected as the sulphonating agent because the reaction is simple and easy and is known to produce polymers free from degradation reactions. Regarding thermal sulfonation, I think you can follow a methodology similar to this study with additional heating. In this case, the reaction under controlled heating will have a direct influence on the depth of treatment on the surface (Kurtz, S. M., PEEK biomaterials handbook, 2019; Gastinel, C.F.J; Kenny, J.M. Plastics Design library, 2017). This study is currently under way.

  • Wang, C.J. Luo, J. Huang, M. Edirisinghe, J. R. Soc. Interface 16 (2019) 20180955. Added on line 55-59.
  • M.H.D. Othman, A. F. Ismail, A. Mustafa, Malaysian Polym. J. 2 (2007) 10 -28. Added on line 64-68.

Results: The virgin PEEK is semi-crystalline with crystallinity ranging from 30 to 35%. After chemical modification on the surface, hydrogen will be replaced by sulfonic acid (SO3H), increasing intermolecular interactions. However, sulfonic acid is bulky, making crystallization difficult. The modified PEEK’s structure still remains semi-crystalline.

Reviewer 2 Report

Honestly, I do not see  the originality and innovation of the presented results. Sulphonation and oxidation of PEEK is known since long time and also the increase in hydrophilicity as consequence. That this increase leads to better cell adhesion (which was not significant here) due to enhanced wetting is common sense. You state that there is only a few literature on influencing parameters as concentration and time. Here, only two different time intervals for the Piranha treatment were stated with not much difference in results.

But nevertheless, it is a well written study and an adequate methodological design. Maybe you can highlight the originality!

Author Response

With respect to the results presented here, the PEEK sulfonation with sulfuric acid and piranha solution has been studied to a very limited extent before (e.g. Stawarczyk, B.; Jordan, P.; Schmidlin, P. R.; Roos, M.; Eichberger, M.; Gernet, W.; Keul, C., The Journal of Prosthetic Dentistry 2014, 112, (5), 1278-1288 and Uhrenbacher, J.; Schmidlin, P. R.; Keul, C.; Eichberger, M.; Roos, M.; Gernet, W.; Stawarczyk, B., J Prosthet Dent 2014, 112, (6), 1489-97, both are cited in this manuscript). In addition to that, biological properties including cell adhesion and proliferation, as well as cytotoxicity assays, were investigated in this study (which were not presented in previous studies).   And about the parameters reported here, these were only part of a thesis study, where other combinations of time and concentration were evaluated, however, this study is still under consideration in a journal.